

# Combined effects of temperature, photoperiod, and salinity on reproduction of the brine shrimp *Artemia sinica* (Crustacea: Anostraca)

Jing-Yu Yang and Shi-Chun Sun

Key Laboratory of Mariculture (Ministry of Education), and Institute of Evolution & Marine Biodiversity, Ocean University of China, Qingdao, Shandong, China

## ABSTRACT

**Background**. *Artemia sinica* is a brine shrimp species distributed in hypersaline salt lakes in northern China and Siberia and a successful invasive species in some coastal salterns. Although it is a commercially harvested and cultured species, knowledge of its reproductive characteristics is limited, and existing studies are often contradictory. The combined effects of temperature, salinity, and photoperiod on reproduction characteristics are experimentally studied to better understand its reproductive features.

**Methods**. There were 36 combinations of three environmental factors ($3 \times 3 \times 4$), each with three or four levels, namely temperature (16, 25, 30 °C), photoperiod (6 L:18 D, 12 L:12D, 18 L:6D), and salinity (50, 100, 150, 200 PSU). In each treatment, 48 to 80 pairs of *A. sinica* from Yuncheng Salt Lake (Shanxi, China) were cultured. Females were observed daily for reproductive mode and the number of offspring produced.

**Results**. Temperature, photoperiod, salinity, and their interactions significantly affected the lifespan and reproduction of *A. sinica*. The reproductive period was the longest and accounted for the largest proportion of life span at moderate temperature (25 °C). Total offspring, offspring per brood, and offspring per day increased as salinity decreased, and the number of broods per female was highest at 25 °C. Temperature, photoperiod, and salinity significantly influenced reproductive modes, and interactions among these factors were identified. *Artemia sinica* primarily reproduces oviparously under low temperature and short daylight conditions, and ovoviviparously under high temperature and long daylight conditions, with the maximum oviparity ratio recorded in treatments of 16 °C, 6L:18D, and 50 or 100 PSU. The maximum ovoviviparity ratio was recorded under 30 °C, 12L:12D, and 100 PSU. Unlike that documented for other *Artemia* species or populations, the brood size of *A. sinica* kept increasing throughout the reproductive period. It did not decline even in the last two broods. For the same brood number, the sizes of oviparous and ovoviviparous broods were similar. The length of the oviparous interval was often greater than that of the ovoviviparous interval, suggesting that oviparous offspring might require additional energy and time to construct the multi-layered eggshell. Compared to other species and populations, the *A. sinica* from Yuncheng Salt Lake has a relatively shorter pre-reproductive development time, a preference for ovoviviparity, and relatively higher fecundity and population growth capacity, making it a suitable culture species for obtaining fresh biomass.

Corresponding author
Shi-Chun Sun, sunsc@ouc.edu.cn

## INTRODUCTION

The brine shrimp *Artemia* (Crustacea: Anostraca) is found on all continents except Antarctica, and its habitats include hypersaline waters, such as inland salt lakes and artificial salt flats (*Van Stappen, 2002*). The genus *Artemia* contains several bisexual species and numerous parthenogenetic populations, both with two reproductive modes: oviparity (producing diapause resting eggs) and ovoviviparity (producing swimming nauplii) (*Liang & MacRae, 1999*). Oviparity, or diapause, is an evolutionary adaptation of *Artemia* to cope with periodic "adverse conditions". Females can control their reproductive modes by sensing impending environmental conditions (*Gajardo & Beardmore, 2012*). Although some *Artemia* populations, such as Hoh Salt Lake (Qinghai, China) (*Bian, 1990*), Caka Salt Lake (Qinghai, China) (*Chang, Asem & Sun, 2017*), and Larnaca Salt Lake (Cyprus) (*Browne & Wanigasekera, 2000*), appear to be "naturally" oviparous, environmental factors play a crucial role in determining the reproductive modes for most *Artemia* species or populations. Studies showed that low temperatures (*Browne, 1980*; *Berthélémy-Okazaki & Hedgecock, 1987*; *Wang, Asem & Sun, 2017*) and short daylight (*Provasoli & Pintner, 1980*; *Huang, Chen & Liu, 2001*; *Nambu, Tanaka & Nambu, 2004*; *Wang, Asem & Sun, 2017*) were usually effective factors for the induction of oviparity. Other factors influencing the reproductive mode include salinity (*Dana & Lenz, 1986*; *Berthélémy-Okazaki & Hedgecock, 1987*; *Jia et al., 1995a*; *Browne & Wanigasekera, 2000*; *Abatzopoulos et al., 2003*; *Sui et al., 2013*; *Aalamifar et al., 2014*), the amount of dissolved oxygen (*Berthélémy-Okazaki & Hedgecock, 1987*), the type and abundance of food (*Browne & Wanigasekera, 2000*), and the addition of chelated iron to the culture medium (*Versichele & Sorgeloos, 1980*). Studies have also shown that there might be interactions between different factors (*Berthélémy-Okazaki & Hedgecock, 1987*; *Wang, Asem & Sun, 2017*).

In addition to the reproductive mode, other reproductive characteristics of *Artemia* can be affected by environmental factors. Several studies showed that the females' lifespan was negatively correlated with temperature (*Browne, Davis & Sallee, 1988*; *Jia et al., 1995b*; *Barata et al., 1996*; *Browne & Wanigasekera, 2000*; *Abatzopoulos et al., 2003*). Under higher salinities, maturity was significantly delayed (*Triantaphyllidis et al., 1995*; *Wang & Zhang, 1995*; *Abatzopoulos et al., 2003*), reproductive output was reduced (*Dana & Lenz, 1986*; *Triantaphyllidis et al., 1995*; *Wang & Zhang, 1995*; *Jia et al., 1995a*; *Sui et al., 2013*), and the post-reproductive period was shortened (*Abatzopoulos et al., 2003*). Maturity and fecundity were also significantly affected by temperature (*Jia et al., 1995b*; *Browne & Wanigasekera, 2000*). *Wear, Haslett & Alexander (1986)* concluded that temperature and salinity significantly affected maturation time, reproductive interval, reproductive period, offspring per brood, and the number of broods of *Artemia*. *Zheng et al. (2019)* found that total offspring, offspring per brood, and the number of broods of *Artemia* increased significantly at higher feeding levels.

*Artemia sinica* Cai, 1989 is mainly distributed in Inner Mongolia, Shanxi, Hebei, and Jilin provinces in China (*Hou, Yang & Cai, 1997*; *Zheng & Sun, 2013*), and is also recorded in Siberia, Russia (*Shadrin & Anufriieva, 2012*; *Litvinenko, Litvinenko & Boyko, 2016*), with Yuncheng Salt Lake (Shanxi, China) probably harboring the largest population. In recent decades, *A. sinica* has spread as an exotic species to several solar salterns in Tianjin and Hebei, China (*Van Stappen et al., 2007*). *Artemia sinica* is an important species, either caught or cultured for fresh biomass (*Jing, 2020*). *Wang et al. (1988)* reported that the *A. sinica* natural population in Yuncheng Salt Lake had 5–6 generations a year and was predominantly oviparous in spring and summer. Since then, several experimental studies have documented the effects of environmental factors on reproduction for this population, but the results were often contradictory. For example, *Wang & Zhang (1995)* found that *A. sinica* was oviparous at 20 °C and 28 °C, and ovoviviparous only at 15 °C, while *Browne & Wanigasekera (2000)* reported only 22.9% of oviparous offspring at 30 °C. *Jia et al. (1995a)* reported that *A. sinica* reproduced successfully at a salinity range of 10–300 PSU and the oviparity proportion increased with salinity. However, *Browne & Wanigasekera (2000)* reported that *A. sinica* did not survive to reproduce under 60 PSU. *Zheng et al. (2019)* reported that *A. sinica* favored oviparity when food was abundant and ovoviviparity when food was insufficient, while *Zhang et al. (2020)* experiments yielded opposite results. Therefore, unknown factors or mechanisms might have influenced the results of these studies. In addition, the effects of photoperiod, the dominant factor in the induction of crustaceans' diapause (*Stross, 1966*; *March, 1982*; *Alekseev, Hwang & Tseng, 2006*; *Wang, Asem & Sun, 2017*), have not been studied in *A. sinica*. To better understand the effects of environmental factors, especially the interactions between different factors on reproductive parameters such as the reproductive mode, various combinations of temperature, photoperiod, and salinity on the reproduction of *A. sinica* were investigated.

## MATERIALS AND METHODS

*Artemia sinica* resting eggs were obtained from the *Artemia* Reference Center, Ghent University (ARC1218). They were collected from Yuncheng Salt Lake, Shanxi, China, in 1991. Nauplii for experiments were obtained by incubating the resting eggs at 25 °C, with continuous light (2000 lx), and a salinity of 30 PSU (natural seawater) for 48 h.

Experiments were conducted at three temperatures (16, 25, and 30 °C), three photoperiods (6 L:18 D, 12 L:12 D, and 18 L:6 D), and four salinities (50, 100, 150, and 200 PSU), resulting in 36 (3 × 3 × 4) treatments. The treatments are denoted as Tx-Lx-Sx (T, temperature; L, light hours; S, salinity; x, value) for narrative convenience. For example, T25-L6-S100 refers to a treatment at 25 °C, a photoperiod of 6 h light:18 h dark, and a salinity of 100 PSU.

High salinity media were prepared by adding sea salt to natural seawater. The experimental conditions at 16 °C and 30 °C were controlled by light incubators (GZH-A250; Shanghai Xianxiang Instruments Co., Ltd., China). The 25 °C experiments were conducted in a constant temperature (25 ± 0.5 °C) incubation chamber with manual control of light hours. The food was a mixture of yeast powder (LANSY-Shrimp ZM powder, INVE Asia

Services Ltd., Thailand) and *Dunaliella* powder (Bioengineering Branch of Inner Mongolia Lantai Industrial Co., Ltd., China). The working suspension was prepared by mixing 1 g of yeast powder, 1 g of *Dunaliella* powder, and 100 g of experimental water by stirring in a mixing cup (MR9800; Guangdong Xinbao Electrical Appliance Co., Ltd., China).

The experiment consisted of two stages. The first stage was performed in 500 ml beakers. Each treatment had three beakers with 200 nauplii and 250 ml of experimental water. Food was supplied 72 h after hatching, approximately 0.005 ml of working suspension per nauplius, and adjusted according to the residue in the following days. During the experiment, all media were exchanged daily. If male and female individuals held pairs, they were transferred to six-well plates, and the experiment entered its second stage. In the second stage, each *Artemia* pair were reared in a well containing about 9 ml of experimental water and 0.1 ml of food suspension. 48–80 *Artemia* pairs were cultured in each treatment, and the remaining males were held in the original beakers as stock cultures. Reproduction events were observed daily, and the reproductive mode and the number of resting eggs or nauplii were recorded. At the same time, all offspring were removed from the cultures, and 100% of the media were exchanged. If the female died, the culture of the pair was terminated, and if the male died, the culture would be continued by adding a new male from the aforementioned stock cultures.

If resting eggs were produced, the offspring were recorded as "oviparous offspring"; if nauplii produced, they were recorded as "ovoviviparous offspring". In most studies, the influence of environmental factors on reproductive mode was assessed by the proportion of resting eggs among the offspring of a female (*Triantaphyllidis et al., 1995*; *Baxevanis et al., 2004*; *Agh et al., 2008*; *Asil et al., 2013*; *Pinto et al., 2014*). However, the reproductive mode (oviparous or ovoviviparous) is mutually exclusive for a given brood of offspring (*Berthélémy-Okazaki & Hedgecock, 1987*), and the percentage of oviparous offspring may be determined primarily by the broods with greater reproductive output (*Wang, Asem & Sun, 2017*). Thus, *Nambu, Tanaka & Nambu (2004)* and *Wang, Asem & Sun (2017)* adopted the percentage of oviparous broods in total broods. In the present study, both of these parameters were calculated.

The following parameters were used in the present study:

Reproductive females: females that produce one or more broods.

Effective females: females that produce two or more broods (*Wang, Asem & Sun, 2017*).

Lifespan (d): days from hatching to death for a reproductive female.

Pre-reproductive period (d): days from hatching to the first reproductive event for a reproductive female.

Reproductive period (d): days from the first to the last reproductive events for an effective female.

Post-reproductive period (d): days from the last reproductive event to death for an effective female.

Reproductive interval (d): days between two adjacent reproductive events for an effective female.

Oviparous interval (d): days between an oviparous event and the previous reproductive event (oviparous or ovoviviparous) for an effective female.

Ovoviviparous interval (d): days between an ovoviviparous event and the previous reproductive event (oviparous or ovoviviparous) for an effective female.

Number of broods: number of reproductive events during the lifetime of a reproductive female.

Total offspring: total number of nauplii and resting eggs a reproductive female produces during the lifetime.

Offspring per brood (brood size): total offspring/number of broods.

Offspring per day: total offspring/lifespan.

Offspring per reproductive day: total offspring/reproductive period.

% Oviparous broods: percentage of oviparous broods in "number of broods" for an effective female.

% Oviparous offspring: percentage of resting eggs in "total offspring" for an effective female.

The effects of salinity, temperature, photoperiod, and their interactions on the observed parameters were assessed by three-way ANOVAs. When significant differences ($p < 0.05$) were detected, a post hoc Turkey test was conducted between treatments. The significance between paired means was analyzed by the F-test. The Pearson simple correlation coefficient between parameters was calculated by the correlation analysis method. All statistical analyses were performed by SPSS 26.

## RESULTS

The percentage of females successfully reproducing was strongly influenced by temperature and salinity. Under lower salinity (50 PSU), both the percentages of successfully reproducing females and the effective females increased with temperature, whereas under higher salinities (150 and 200 PSU), females reproduced only in 25 °C treatments and the T16-L6-S150 treatment (Table 1).

### Lifespan

The lifespan, pre-reproductive period, and post-reproductive period of females were significantly influenced by temperature and salinity, and there were interactions among factors (Table S1). The three parameters were all shortened with increasing temperatures (Table 2). The reproductive periods were the shortest at 30 °C (13.2 ± 6.4 to 18.7 ± 9.0 d), followed by 16 °C (16.9 ± 6.7 to 34.8 ± 15.1 d), and the longest at 25 °C (28.7 ± 15.8 to 42.4 ± 20.2 d) (Table 2). The percentages of the reproductive period in lifespan were the highest at 25 °C (47% to 66%), followed by 30 °C (40% to 46%), and the lowest at 16 °C (19% to 38%) (Fig. 1). That is, the reproductive period was the longest and accounted for the highest proportion of lifespan at the moderate temperature (25 °C).

### Fecundity

Temperature and salinity significantly and independently impacted the reproductive interval (Table S1). The reproductive interval shortened with rising temperatures and falling salinities (Table 2). The maximum interval was 10.7 ± 2.8 d in the T16-L6-S100 treatment, and the minimum was 3.6 ± 0.3 d in the T30-L18-S50 treatment. Except
**Table 1  Numbers of cultured females, reproductive females (≥1 broods), and effective females (≥2 broods) under 36 temperature-photoperiod-salinity treatments.** Percentages of reproductive and effective females in the cultured females are shown in parentheses.

| Conditions | | Cultured females | | | Reproductive females | | | Effective females | | |
|---|---|---|---|---|---|---|---|---|---|---|
| | | L6 | L12 | L18 | L6 | L12 | L18 | L6 | L12 | L18 |
| T16 | S50 | 48 | 48 | 48 | 28 (58.3) | 27 (56.3) | 40 (83.3) | 17 (35.4) | 25 (52.1) | 36 (75.0) |
| | S100 | 48 | 48 | 48 | 47 (97.9) | 41 (85.4) | 42 (87.5) | 41 (85.4) | 29 (60.4) | 31 (64.6) |
| | S150 | 48 | 48 | 48 | 5 (10.4) | 0 (0.0) | 0 (0.0) | 4 (8.3) | 0 (0.0) | 0 (0.0) |
| | S200 | 48 | 48 | 48 | 0 (0.0) | 0 (0.0) | 0 (0.0) | 0 (0.0) | 0 (0.0) | 0 (0.0) |
| T25 | S50 | 60 | 60 | 60 | 40 (66.7) | 43 (71.7) | 49 (81.7) | 38 (63.3) | 40 (66.7) | 47 (78.3) |
| | S100 | 80 | 60 | 72 | 61 (76.3) | 59 (98.3) | 69 (95.8) | 46 (57.5) | 56 (93.3) | 61 (84.7) |
| | S150 | 60 | 60 | 60 | 58 (96.7) | 55 (91.7) | 58 (96.7) | 52 (86.7) | 43 (71.7) | 53 (88.3) |
| | S200 | 60 | 60 | 60 | 15 (25.0) | 12 (20.0) | 22 (36.7) | 12 (20.0) | 10 (16.7) | 12 (20.0) |
| T30 | S50 | 60 | 60 | 60 | 56 (93.3) | 59 (98.3) | 56 (93.3) | 47 (78.3) | 55 (91.7) | 51 (85.0) |
| | S100 | 60 | 60 | 60 | 60(100.0) | 60(100.0) | 59 (98.3) | 53 (88.3) | 53 (88.3) | 55 (91.7) |
| | S150 | 60 | 60 | 60 | 0 (0.0) | 0 (0.0) | 0 (0.0) | 0 (0.0) | 0 (0.0) | 0 (0.0) |
| | S200 | 60 | 60 | 60 | 0 (0.0) | 0 (0.0) | 0 (0.0) | 0 (0.0) | 0 (0.0) | 0 (0.0) |

for the T30-L18-S100 treatment, the oviparous intervals were slightly longer than the ovoviviparous intervals, though the differences were not significant for some treatments (Fig. 2).

The number of broods was significantly influenced by temperature and salinity, and there were interactions among temperature, salinity, and photoperiod (Table S1). Higher number ($4.4 \pm 3.5$ to $9.0 \pm 4.4$) of broods was observed at 25 °C, where two females in two L18 treatments produced the maximum number (20) of broods during their lifetime (127 d, 133 d). The number of broods at 30 °C ($3.7 \pm 0.4$ to $4.8 \pm 2.6$ broods) was slightly higher than at 16 °C ($2.1 \pm 0.9$ to $4.0 \pm 1.7$ broods) (Table 2). The differences in the number of broods among L-S treatments were more pronounced at 25 °C than at 16 °C and 30 °C.

The total offspring, offspring per brood, offspring per day, and offspring per reproductive day were significantly influenced by temperature and salinity, and there were interactions among factors (Table S1). They all decreased with the elevation of salinity (Table 2). The maximum value of total offspring was recorded in the T25-L18-S50 treatment ($587.7 \pm 418.0$), and the minimum value in the T16-L18-S100 treatment ($112.3 \pm 83.6$). Offspring per brood was the highest ($100.8 \pm 45.4$) in the T16-L18-S50 treatment and the lowest ($34.5 \pm 16.9$) in the T25-L12-S150 treatment. Offspring per day and offspring per reproductive day also increased with temperature (Table 2), being maximum ($12.3 \pm 6.0$ and $33.5 \pm 8.7$, respectively) in the T30-L12-S50 treatment and minimum ($1.3 \pm 1.0$ and $8.3 \pm 3.5$, respectively) in the T16-L18-S100 treatment.

## Reproductive mode

The percentages of oviparous broods and oviparous offspring were significantly influenced by temperature, photoperiod, salinity, and their interactions (Table S1). At 16 °C, higher percentages of oviparous broods were observed under shorter daylight and lower salinity conditions, with the highest value ($88.7 \pm 26.1\%$) recorded in the T16-L6-S50 treatment (Table 2). At 25 °C, all treatments had low percentages ($21.3 \pm 21.4\%$ to $56.1 \pm 31.3\%$).

**Table 2  The lifespan and reproductive parameters of *Artemia sinica*.** Data shown as mean ± SD (range). Values with different superscripts in the same column indicate significant differences ($p < 0.05$).

| T (°C) | L (h) | S (PSU) | Lifespan (d) | Pre-reproductive period (d) | Reproductive period (d) | Post-reproductive period (d) | Reproductive interval (d) | Number of broods |
|---|---|---|---|---|---|---|---|---|
| 16 | 6 | 50 | 71.9 ± 17.8[c,d,e] (39~115) | 48.3 ± 6.4[d,e] (37~68) | 28.9 ± 15.0[a,b,c,d,e,f] (5~61) | 5.8 ± 5.9[a,b] (0~21) | 10.5 ± 1.7[a] (9.0~14.0) | 2.8 ± 1.8[f] (1~7) |
| | | 100 | 76.6 ± 16.2[b,c,d] (40~99) | 46.6 ± 8.0[e] (37~68) | 28.2 ± 13.4[a,b,c,d,e,f] (7~53) | 5.4 ± 3.9[a,b,c] (1~15) | 10.8 ± 2.8[a] (6.0~22.0) | 3.3 ± 1.6[e,f] (1~6) |
| | | 150 | 92.2 ± 1.1[a] (91~93) | 77.2 ± 9.9[a] (60~84) | 13.5 ± 6.4[f,g] (10~23) | 3.0 ± 3.4[a,b,c,d] (1~8) | 9.8 ± 1.0[a] (9.0~11.0) | 2.0 ± 0.7[f] (2~3) |
| | 12 | 50 | 87.9 ± 14.1[a,b] (54~111) | 56.0 ± 6.6[c] (44~68) | 33.4 ± 11.9[a,b,c] (13~59) | 4.9 ± 3.1[a,b,c,d] (0~12) | 10.0 ± 0.9[a] (9.0~13.3) | 3.6 ± 1.3[d,e,f] (1~6) |
| | | 100 | 84.9 ± 10.5[a,b,c] (50~102) | 64.8 ± 10.8[b] (45~84) | 19.9 ± 10.2[b,c,d,e,f,g] (9~45) | 4.9 ± 3.6[a,b,c,d] (1~13) | 10.1 ± 0.8[a] (8.0~12.0) | 2.3 ± 1.2[f] (1~5) |
| | 18 | 50 | 89.2 ± 17.7[a,b] (47~113) | 52.5 ± 6.3[c,d] (45~69) | 34.8 ± 15.1[a,b] (8~60) | 5.4 ± 2.7[a,b,c] (1~11) | 10.1 ± 1.7[a] (7.0~17.0) | 4.0 ± 1.7[d,e,f] (1~7) |
| | | 100 | 85.2 ± 7.1[a,b,c] (65~96) | 66.9 ± 7.8[b] (51~87) | 16.9 ± 6.7[d,e,f,g] (9~31) | 5.1 ± 3.8[a,b,c,d] (1~14) | 10.7 ± 2.7[a] (8.0~20.0) | 2.1 ± 0.9[f] (1~4) |
| 25 | 6 | 50 | 59.8 ± 15.0[e,f] (29~94) | 29.7 ± 8.2[g] (18~46) | 28.7 ± 15.8[a,b,c,d,e,f] (5~69) | 2.9 ± 2.1[a,b,c,d] (0~9) | 5.0 ± 0.4[b,c,d,e] (4.0~6.0) | 6.3 ± 3.1[a,b,c,d] (1~13) |
| | | 100 | 51.5 ± 19.9[f,g] (23~95) | 23.7 ± 5.0[h,i,j] (19~44) | 28.8 ± 18.6[a,b,c,d,e,f] (6~74) | 5.9 ± 4.7[a,b] (0~17) | 5.6 ± 0.9[b] (4.5~10.0) | 4.4 ± 3.5[c,d,e,f] (1~15) |
| | | 150 | 58.9 ± 21.0[e,f] (29~107) | 25.8 ± 4.2[g,h] (22~51) | 32.7 ± 20.8[a,b,c,d] (5~78) | 3.6 ± 2.6[a,b,c,d] (0~16) | 5.7 ± 3.5[b] (4.0~30.0) | 6.2 ± 4.1[a,b,c,d,e] (1~15) |
| | | 200 | 51.1 ± 13.3[f,g] (34~80) | 36.0 ± 4.3[f] (30~44) | 13.8 ± 9.9[e,f,g] (3~32) | 4.4 ± 9.1[a,b,c,d] (0~33) | 4.9 ± 1.0[b,c,d,e] (2.0~6.0) | 3.0 ± 1.9[f] (1~7) |
| | 12 | 50 | 60.1 ± 18.3[e,f] (26~103) | 26.6 ± 7.0[g,h] (16~39) | 33.0 ± 16.9[a,b,c,d] (5~70) | 2.5 ± 1.7[a,b,c,d] (0~6) | 4.7 ± 0.3[b,c,d,e] (3.5~5.1) | 7.0 ± 3.6[a,b,c] (1~14) |
| | | 100 | 63.2 ± 19.4[d,e,f] (25~107) | 19.7 ± 2.3[i,j,k] (16~29) | 41.3 ± 18.4[a] (10~87) | 4.1 ± 4.2[a,b,c,d] (0~21) | 5.2 ± 0.5[b,c] (4.3~6.8) | 8.2 ± 4.0[a] (1~18) |
| | | 150 | 53.9 ± 23.4[f] (24~119) | 25.3 ± 4.0[g,h] (21~38) | 30.0 ± 22.5[a,b,c,d,e] (6~87) | 5.3 ± 6.6[a,b,c] (0~35) | 5.4 ± 0.6[b] (4.3~7.0) | 4.9 ± 4.1[b,c,d,e,f] (1~17 |
| | | 200 | 48.2 ± 7.5[f,g] (40~65) | 36.8 ± 2.7[f] (33~41) | 11.2 ± 7.6[g] (5~28) | 2.0 ± 1.6[c,d] (1~6) | 5.8 ± 0.7[b,c,d,e] (4.0~6.0) | 2.7 ± 1.4[f] (1~6) |
| | 18 | 50 | 62.3 ± 21.3[d,e,f] (23~105) | 19.0 ± 3.3[k] (16~33) | 42.4 ± 20.2[a] (6~84) | 2.6 ± 1.7[a,b,c,d] (0~8) | 4.9 ± 0.5[b,c,d,e] (4.3~6.3) | 9.0 ± 4.4[a] (1~19) |
| | | 100 | 57.9 ± 24.7[e,f] (22~127) | 19.6 ± 1.2[i,j,k] (18~22) | 39.2 ± 23.2[a] (4~104) | 3.5 ± 2.8[a,b,c,d] (0~13) | 5.1 ± 0.7[b,c,d] (3.0~7.0) | 7.4 ± 4.8[a,b] (1~20) |
| | | 150 | 55.7 ± 23.5[f] (25~133) | 24.1 ± 2.6[h,i] (19~34) | 31.2 ± 23.4[a,b,c,d] (3~103) | 2.9 ± 3.0[a,b,c,d] (0~12) | 5.1 ± 0.7[b,c] (2.0~7.5) | 6.1 ± 4.4[a,b,c,d,e] (1~20) |
| | | 200 | 51.4 ± 12.4[f,g] (35~72) | 35.4 ± 4.0[f] (29~47) | 18.4 ± 8.8[b,c,d,e,f,g] (3~36) | 6.1 ± 4.3[a] (0~15) | 5.7 ± 1.3[b] (2.0~7.0) | 2.5 ± 1.7[f] (1~6) |
| 30 | 6 | 50 | 31.5 ± 7.3[h] (20~46) | 18.3 ± 2.6[k] (16~33) | 13.2 ± 6.4[f,g] (4~25) | 1.7 ± 1.5[d] (0~5) | 3.8 ± 0.4[e] (2.9~5.5) | 3.6 ± 1.9[d,e,f] (1~7) |
| | | 100 | 38.3 ± 10.5[g,h] (21~58) | 19.3 ± 1.2[j,k] (17~23) | 18.7 ± 9.0[b,c,d,e,f,g] (4~36) | 2.4 ± 1.8[b,c,d] (0~9) | 3.9 ± 0.5[d,e] (2.5~5.5) | 4.8 ± 2.6[b,c,d,e,f] (1~11) |
| | 12 | 50 | 31.0 ± 8.6[h] (17~48) | 16.3 ± 1.9[k] (14~24) | 13.3 ± 7.8[f,g] (4~30) | 2.4 ± 2.0[b,c,d] (0~13) | 3.7 ± 0.4[e] (2.6~5.0) | 4.1 ± 2.2[c,d,e,f] (1~9) |

**Table 2** (*continued*)

| T (°C) | L (h) | S (PSU) | Lifespan (d) | Pre-reproductive period (d) | Reproductive period (d) | Post-reproductive period (d) | Reproductive interval (d) | Number of broods |
|---|---|---|---|---|---|---|---|---|
| | | 100 | 38.2 ± 8.1[g,h] (21~56) | 18.9 ± 1.3[k] (16~23) | 18.0 ± 6.8[c,d,e,f,g] (4~33) | 3.2 ± 3.8[a,b,c,d] (0~18) | 3.9 ± 0.5[c,d,e] (3.0~6.7) | 4.6 ± 2.1[b,c,d,e,f] (1~8) |
| | 18 | 50 | 31.0 ± 7.7[h] (18~49) | 16.1 ± 1.8[k] (13~22) | 14.2 ± 7.1[e,f,g] (4~32) | 1.8 ± 1.5[d] (0~5) | 3.6 ± 0.3[e] (2.0~5.0) | 4.3 ± 2.1[c,d,e,f] (1~9) |
| | | 100 | 36.7 ± 11.8[g,h] (19~58) | 18.9 ± 2.1[k] (14~25) | 16.8 ± 10.5[d,e,f,g] (3~36) | 2.2 ± 2.4[c,d] (0~11) | 3.8 ± 0.8[d,e] (2.0~7.0) | 4.8 ± 2.7[b,c,d,e,f] (1~10) |

| T (°C) | L (h) | S (PSU) | Total offspring | Offspring per brood | Offspring per day | Offspring per reproductive day | % Oviparous broods | % Oviparous offspring |
|---|---|---|---|---|---|---|---|---|
| 16 | 6 | 50 | 240.2 ± 277.5[c,d,e,f,g,h] (5~1048) | 64.9 ± 43.1[c,d,e] (5.0~149.7) | 2.8 ± 2.7[h,i,j] (0.1~9.1) | 12.8 ± 7.0[e,f,g] (3.7~32.2) | 88.7 ± 26.1[a] (0.0~100.0) | 87.9 ± 28.3[a] (0.0~100.0) |
| | | 100 | 206.5 ± 147.8[d,e,f,g,h] (16~597) | 57.6 ± 23.5[c,d,e,f] (9.0~110.0) | 2.5 ± 1.5[h,i,j] (0.2~6.7) | 8.3 ± 3.5[f,g,h] (1.5~16.5) | 87.0 ± 26.8[a] (0.0~100.0) | 85.7 ± 28.7[a,b] (0.0~100.0) |
| | | 150 | 81.6 ± 20.6[g,h] (66~117) | 46.7 ± 23.3[e,f,g,h] (22.0~81.0) | 0.9 ± 0.2[j] (0.7~1.3) | 7.0 ± 3.2[g,h] (2.9~10.6) | 25.0 ± 50.0[g,h,I,i] (0.0~100.0) | 25.0 ± 50.0[f,g,h,I,j] (0.0~100.0) |
| | 12 | 50 | 365.3 ± 221.9[a,b,c,d,e,f] (19~865) | 95.3 ± 37.9[a,b] (19.0~191.0) | 3.9 ± 2.0[f,g,h,i,j] (0.2~8.2) | 11.3 ± 3.9[e,f,g,h] (5.9~21.8) | 73.4 ± 32.8[a,b,c] (0.0~100.0) | 77.4 ± 31.7[a,b,c] (0.0~100.0) |
| | | 100 | 133.0 ± 114.6[e,f,g,h] (4~410) | 49.9 ± 26.1[d,e,f,g,h] (4.0~110.3) | 1.5 ± 1.2[i,j] (0.0~4.4) | 8.8 ± 3.8[f,g,h] (2.2~15.2) | 45.0 ± 42.1[c,d,e,f,g,h] (0.0~100.0) | 48.8 ± 43.5[c,d,e,f,g,h] (0.0~100.0) |
| | 18 | 50 | 457.7 ± 314.1[a,b,c,d] (24~1047) | 100.8 ± 45.4[a] (24.00~187.0) | 4.8 ± 2.9[d,e,f,g,h,i] (0.3~10.4) | 14.1 ± 4.7[e,f] (4.9~23.3) | 31.8 ± 30.7[e,f,g,h,i] (0.0~100.0) | 29.7 ± 32.4[e,f,g,h,i,j] (0.0~100.0) |
| | | 100 | 112.3 ± 83.6[f,g,h] (10~354) | 48.8 ± 22.4[e,f,g,h] (10.0~95.00) | 1.3 ± 1.0[i,j] (0.1~4.1) | 8.3 ± 3.5[f,g,h] (1.7~15.8) | 20.2 ± 32.8[h,i] (0.0~100.0) | 17.1 ± 31.6[h,i,j] (0.0~100.0) |
| 25 | 6 | 50 | 385.1 ± 320.3[a,b,c,d,e,f] (35~1336) | 54.3 ± 22.7[c,d,e,f,g] (20.5~108.6) | 5.8 ± 3.6[c,d,e,f,g,h] (0.9~15.3) | 12.9 ± 4.6[e,f,g] (6.0~25.5) | 54.4 ± 33.5[b,c,d,e,f,g] (0.0~100.0) | 54.2 ± 36.1[b,c,d,e,f,g] (0.0~100.0) |
| | | 100 | 212.9 ± 292.4[d,e,f,g,h] (10~1585) | 40.8 ± 18.6[f,g,h,i] (10.0~113.2) | 3.4 ± 3.1[g,h,i,j] (0.4~17.0) | 8.9 ± 4.6[f,g,h] (2.1~24.8) | 31.4 ± 32.4[e,f,g,h,i] (0.0~100.0) | 31.3 ± 33.1[e,f,g,h,i,j] (0.00~100.00) |
| | | 150 | 294.8 ± 283.2[b,c,d,e,f,g,h] (1~1095) | 39.9 ± 17.4[f,g,h,i] (1.0~84.2) | 4.2 ± 3.0[e,f,g,h,i,j] (0.0~11.8) | 9.1 ± 3.8[f,g,h] (1.6~16.6) | 33.4 ± 28.9[f,g,h,i] (0.0~100.0) | 32.8 ± 30.7[e,f,g,h,i,j] (0.0~100.0) |
| | | 200 | 112.9 ± 102.1[f,g,h] (22 ~345) | 33.8 ± 12.0[g,h,i] (15.5~57.5) | 2.1 ± 1.5[i,j] (0.4~5.8) | 10.3 ± 4.8[e,f,g,h] (4.4~20.7) | 29.8 ± 34.6[f,g,h,i] (0.0~100.0) | 29.8 ± 34.8[e,f,g,h,i,j] (0.0~100.0) |
| | 12 | 50 | 532.9 ± 422.4[a,b] (7~1899) | 65.9 ± 28.5[c,d,e] (7.0~135.6) | 7.7 ± 4.5[b,c,d,e] (0.2~18.4) | 16.2 ± 5.9[d,e] (4.9~29.8) | 41.6 ± 31.6[d,e,f,g,h] (0.0~100.0) | 44.7 ± 33.2[d,e,f,g,h,i] (0.0~100.0) |
| | | 100 | 515.1 ± 402.1[a,b,c] (38~1780) | 56.9 ± 20.8[c,d,e,f,g] (17.7~113.2) | 7.4 ± 4.1[b,c,d,e,f] (1.0~18.2) | 12.2 ± 5.0[e,f,g] (4.1~25.4) | 21.3 ± 21.4[h,i] (0.0~100.0) | 22.8 ± 24.6[g,h,I,j] (0.0~100.0) |
| | | 150 | 213.5 ± 267.8[d,e,f,g,h] (5~1102) | 34.5 ± 16.9[f,g,h,i] (5.0~81.9) | 3.1 ± 2.7[g,h,i,j] (0.2~12.4) | 8.2 ± 3.3[f,g,h] (2.4~17.6) | 26.0 ± 30.5[f,g,h,i] (0.0~100.0) | 25.9 ± 32.6[f,g,h,i,j] (0.0~100.0) |
| | | 200 | 79.0 ± 75.0[g,h] (13~277) | 26.4 ± 10.6[h,i] (13.0~46.2) | 1.5 ± 1.1[i,j] (0.3~4.3) | 7.8 ± 1.8[f,g,h] (5.7~10.3) | 29.2 ± 24.6[f,g,h,i] (0.0~100.0) | 29.5 ± 25.4[e,f,g,h,i,j] (0.0~100.0) |
| | 18 | 50 | 587.7 ± 417.9[a] (1~1789) | 57.9 ± 22.7[c,d,e,f] (1.0~97.9) | 8.3 ± 4.2[b,c,d] (0.0~17.0) | 13.4 ± 4.4[e,f] (1.5~21.3) | 31.5 ± 21.2[e,f,g,h,i] (0.0~100.0) | 36.5 ± 26.9[e,f,g,h,i,j] (0.00~100.00) |
| | | 100 | 474.0 ± 526.8[a,b,c,d] (4~2698) | 51.2 ± 26.5[d,e,f,g] (4.0~134.9) | 6.5 ± 4.8[c,d,e,f,g] (0.2~21.4) | 11.7 ± 5.3[e,f,g] (4.0~27.1) | 31.9 ± 29.7[e,f,g,h,i] (0.0~100.0) | 33.2 ± 32.3[f,g,h,i,j] (0.0~100.0) |
| | | 150 | 299.6 ± 354.6[b,c,d,e,f,g,h] (12~1633) | 38.0 ± 18.9[f,g,h,i] (9.3~84.3) | 4.2 ± 3.4[e,f,g,h,i,j] (0.3~12.3) | 9.0 ± 4.0[f,g,h] (2.6~20.6) | 56.1 ± 31.3[b,c,d,e,f] (0.0~100.0) | 56.6 ± 32.6[a,b,c,d,e,f] (0.0~100.0) |
| | | 200 | 51.6 ± 45.1[h] (4~183) | 19.0 ± 8.4[i] (4.0~32.3) | 0.9 ± 0.7[j] (0.1~2.5) | 5.3 ± 3.7[h] (1.9~16.0) | 60.4 ± 40.0[a,b,c,d,e] (0.0~100.0) | 59.5 ± 41.9[a,b,c,d,e] (0.0~100.0) |

**Table 2** (*continued*)

| T (°C) | L (h) | S (PSU) | Total offspring | Offspring per brood | Offspring per day | Offspring per reproductive day | % Oviparous broods | % Oviparous offspring |
|---|---|---|---|---|---|---|---|---|
| 30 | 6 | 50 | 349.1 ± 220.8[a,b,c,d,e,f,g] (32~923) | 89.9 ± 26.8[a,b] (32.0~143.7) | 10.4 ± 5.2[a,b] (1.0~20.5) | 32.5 ± 9.2[a,b] (9.2~53.9) | 24.4 ± 30.8[g,h,i] (0.0~100.0) | 23.5 ± 29.8[g,h,I,j] (0.0~100.0) |
| | | 100 | 355.9 ± 277.6[a,b,c,d,e,f,g] (21~1212) | 65.8 ± 25.4[c,d,e] (21.0~155.4) | 8.3 ± 5.0[b,c,d] (0.8~21.3) | 20.5 ± 7.3[c,d] (5.9~38.9) | 15.7 ± 26.8[h,i] (0.0~100.0) | 16.4 ± 28.8[i,j] (0.00~100.00) |
| | 12 | 50 | 415.3 ± 280.2[a,b,c,d,e] (38~1035) | 95.5 ± 27.6[a,b] (31.0~162.5) | 12.3 ± 6.0[a] (1.8~22.9) | 33.5 ± 8.7[a] (10.3~54.2) | 18.8 ± 21.8[h,i] (0.0~100.0) | 21.6 ± 25.5[h,i,j] (0.0~78.2) |
| | | 100 | 368.5 ± 239.0[a,b,c,d,e,f] (8~922) | 72.9 ± 27.7[b,c,d] (8.0~153.7) | 9.2 ± 5.2[a,b,c] (0.2~24.3) | 24.0 ± 11.2[c] (1.0~67.8) | 4.5 ± 12.4[i] (0.0~100.0) | 4.7 ± 14.5[j] (0.0~65.79) |
| | 18 | 50 | 345.1 ± 205.5[a,b,c,d,e,f,g] (27~922) | 77.6 ± 21.0[a,b,c] (27.0~132.0) | 10.4 ± 4.5[a,b] (1.2~21.9) | 26.6 ± 6.3[b,c] (14.0~44.0) | 68.7 ± 26.1[a,b,c,d] (0.0~100.0) | 69.3 ± 27.3[a,b,c,d] (0.0~100.0) |
| | | 100 | 250.0 ± 208.3[b,c,d,e,f,g,h] (21~932) | 47.7 ± 19.0[e,f,g,h] (21.0~116.5) | 6.1 ± 3.8[c,d,e,f,g,h] (0.9~18.6) | 16.3 ± 6.0[d,e] (5.3~32.1) | 76.8 ± 30.6[a,b] (0.0~100.0) | 76.8 ± 31.0[a,b,c,d] (0.0~100.0) |

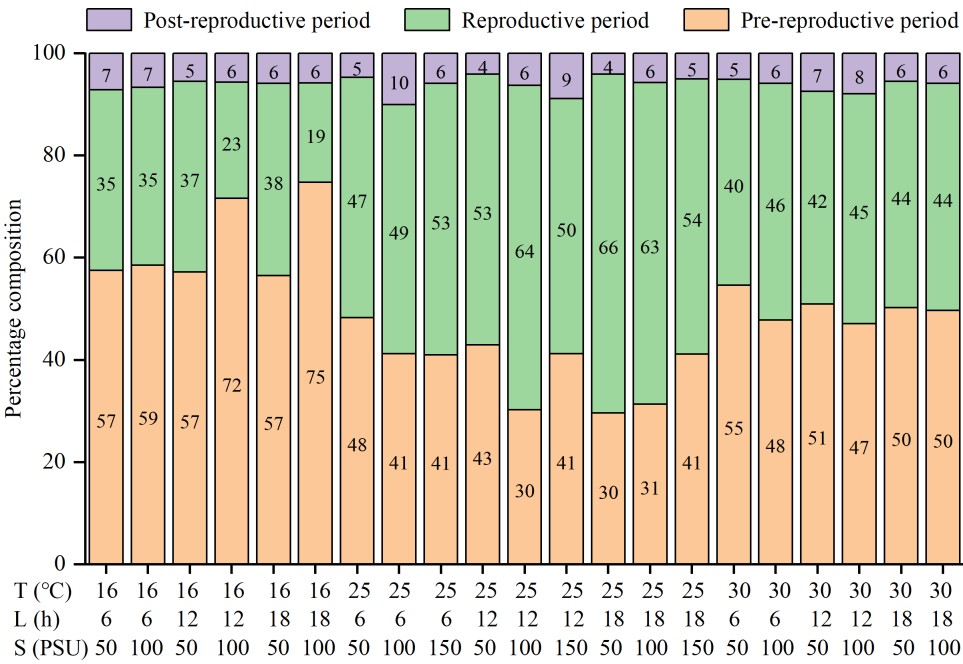

**Figure 1** Percentage composition of pre-reproductive period, reproductive period and post-reproductive period in lifespan of female *Artemia sinica*.

Interestingly, at 30 °C, the L18 treatments had much higher percentages (about 70%) than the L6 and L12 treatments (4.5 ± 12.4% to 24.4 ± 30.8%), with the minimum recorded value in the T30-L12-S100 treatment. The trend in the percentage of oviparous offspring was highly consistent with the percentage of oviparous broods (Table 2). Among the 25 treatments, the percentages of oviparous broods were greater than 50% in seven treatments, greater than 70% in four treatments, and greater than 80% in two treatments, indicating a preference for ovoviviparity. In addition, some females were always ovoviviparous while some others were always oviparous throughout their lifetime, and their proportions varied

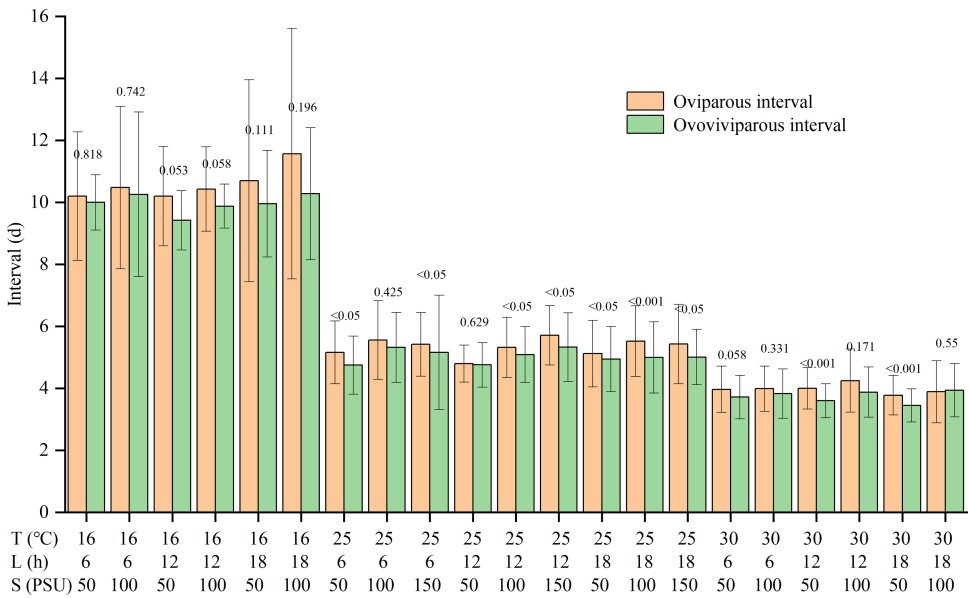

**Figure 2  Oviparous and ovoviviparous intervals of *Artemia sinica*.** The number above two bars of each treatment is the *p* value of the F-test.

with the rearing condition (Fig. 3). For example, in the T16-L6-S50 treatment, 73% of females were constantly oviparous, and no females were exclusively ovoviviparous; in the T30-L12-S100 treatment, 85% of females were constantly ovoviviparous, and no females were exclusively oviparous; in the T16-L18-S100 treatment, all females produced both oviparous and ovoviviparous broods.

## Variation of reproductive parameters concerning brood number

Figure 4 shows the variations in the percentage of oviparity, brood size, and reproductive interval in relation to brood number (results were grouped by temperature because the numbers of broods were strongly influenced by temperature). The oviparity percentage curves often fluctuated greatly (Figs. 4A–4C). At 16 °C, the L6-S50, L12-S50, and L12-S100 curves decreased first and then increased, while the L6-S100, L18-S50, and L18-S100 curves always decreased with brood number (Fig. 4A). At 25 °C, curves tended to fluctuate downward (Fig. 4B). At 30 °C, the general tendency of oviparity was to decline with brood number, which was particularly pronounced in the L18-S50 and L18-S100 treatments. In most treatments, the oviparity percentage in the first brood was somewhat lower than in the second brood (Fig. 4C).

In all treatments, the number of offspring per brood increased with brood number, which was more pronounced in 16 and 25 °C treatments than in 30 °C treatments (Figs. 4D–4F). The offspring of the last brood was 1.95–4.78 times as many as those of the first brood at 16 °C, 1.47–5.52 times at 25 °C, and 1.44–2.22 times at 30 °C (Table 3). The brood size comparison of the last three broods showed that the *A. sinica* brood size remained increasing even near the end of the reproductive period (the ratios of a later brood to the

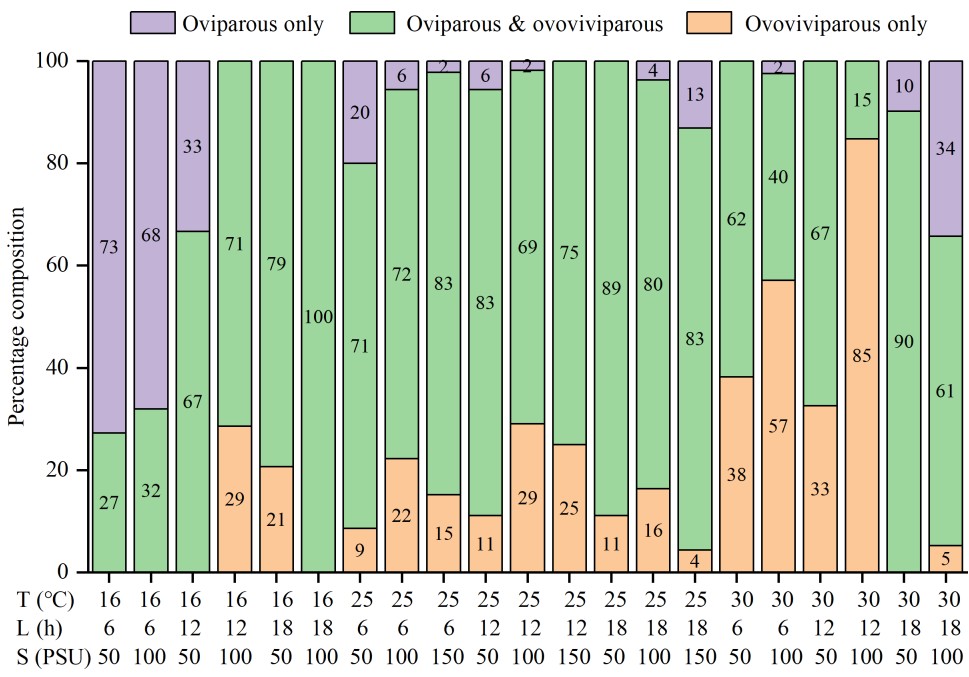

**Figure 3** Percentage composition of females that the reproductive mode was oviparous only, ovoviviparous only, and both oviparous and ovoviviparous under various conditions.

adjacent previous brood were generally greater than 1; Table 3). Moreover, for the same brood number, the oviparous and ovoviviparous brood sizes did not show any noticeable difference (their ratios were mostly close to 1; Table S2).

The reproductive interval did not vary along with brood number (Figs. 4G–4I). At 16 °C, the fluctuation in the L18-S50 treatment was within ±2 d, and those in other treatments were within ±1 d (Fig. 4G). At 25 °C, all fluctuations were within ±1 d (Fig. 4H). At 30 °C, all fluctuations were only within ±0.5 d (Fig. 4I).

### Linear correlation between lifespan and reproductive parameters

Data from the T25-L12-S100 treatment was analyzed and is shown in Table S3. There were strong positive correlations: (1) among lifespan, reproductive period, number of broods, and total offspring; (2) among total offspring, offspring per brood, offspring per day, and offspring per reproductive day; and (3) between the percentage of oviparous broods and the percentage of oviparous offspring. The pre-reproductive period, post-reproductive period, and reproductive interval were not significantly correlated with other parameters.

## DISCUSSION

### Reproductive mode

The three-way ANOVAs showed that the percentage of oviparous broods and the percentage of oviparous offspring were influenced by temperature, photoperiod, and salinity; photoperiod–salinity, photoperiod–temperature, and temperature–salinity had interactions; but temperature–photoperiod–salinity did not show significant interaction

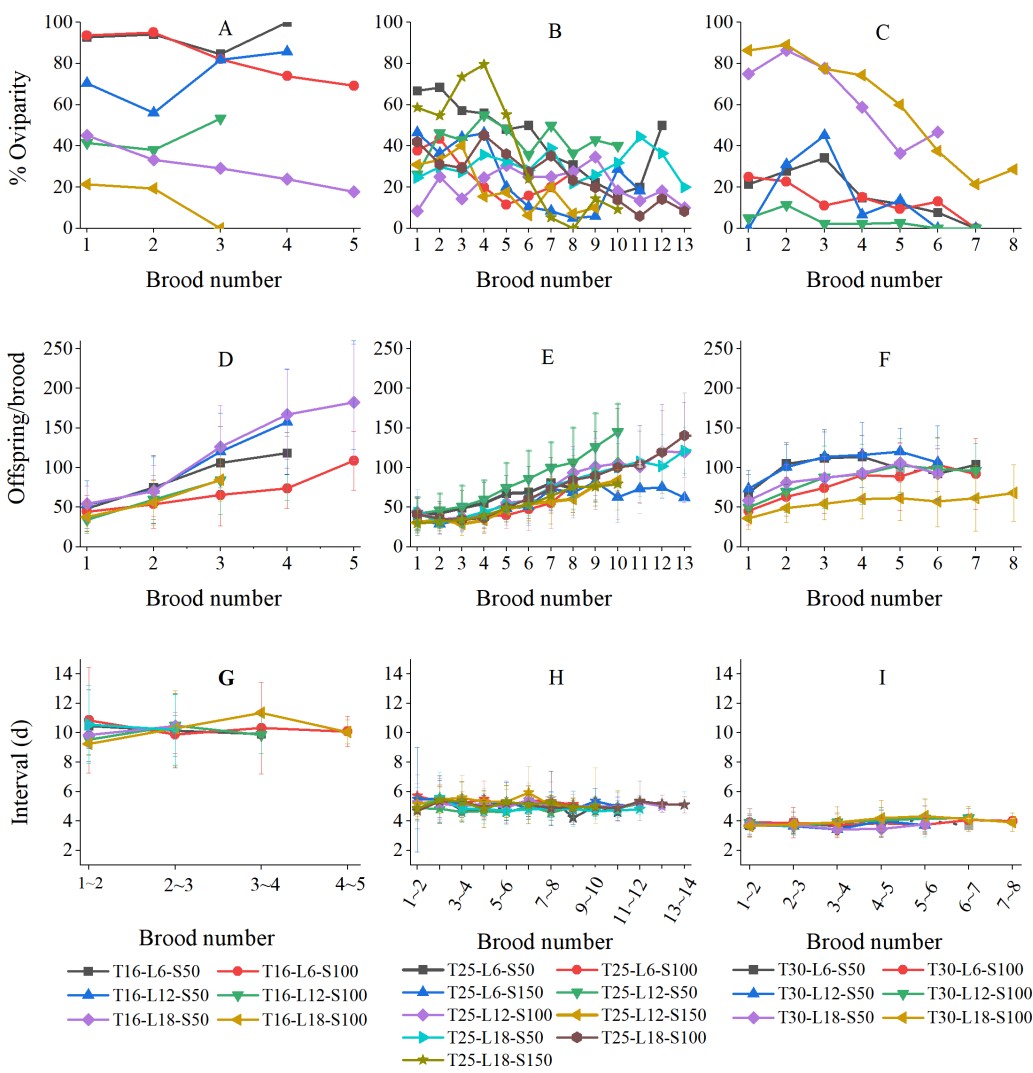

**Figure 4** **Variation of reproductive parameters of *Artemia sinica* in relation to brood number.** (A–C) Percentage of oviparity. (D–F) Offspring per brood. (G–I) Reproductive interval. Results are grouped by temperatures (left column, 16 °C treatments; middle column, 25 °C treatments; right column, 30 °C treatments). Only data with ≥ 10 females are shown.

(Table S1). Few studies documented the combined effects of environmental factors on the reproductive mode of *Artemia*. *Wang, Asem & Sun (2017)* reported interactive effects of temperature, photoperiod, and salinity on the reproductive mode of the parthenogenetic *Artemia* (a 2n clone) from Barkol Salt Lake (Xinjiang, China). Similar to the present study, their results showed that lower temperatures and shorter daylight induce the production of resting eggs, while salinity did not have a significant independent effect on reproductive mode. *Berthélémy-Okazaki & Hedgecock (1987)* found that photoperiod–temperature and temperature–salinity had interactions affecting the reproductive mode of *Artemia* from San Francisco Bay (California, USA), but did not detect a significant interaction between photoperiod and salinity.

**Table 3  Comparison of the brood sizes in the first and last three broods of *Artemia sinica* (ratios of the brood sizes in selected brood numbers).** Data shown as mean ± s.d., which was calculated from values of each female. Only females producing ≥4 broods were included in this analysis. $B_1$, offspring in the first brood; $B_{L1}$, offspring in the last brood; $B_{L2}$, offspring in the second brood from the bottom; $B_{L3}$, offspring in the third brood from the bottom.

| T | L | S | $B_{L1}/B_1$ | $B_{L1}/B_{L2}$ | $B_{L2}/B_{L3}$ |
|---|---|---|---|---|---|
| 16 | 6 | 50 | 1.95 ± 0.99 | 1.45 ± 1.48 | 1.32 ± 0.54 |
| | | 100 | 2.93 ± 4.52 | 1.37 ± 0.89 | 1.59 ± 1.41 |
| | 12 | 50 | 4.78 ± 4.88 | 1.10 ± 0.35 | 1.65 ± 0.87 |
| | | 100 | 3.79 ± 1.92 | 1.17 ± 0.37 | 1.66 ± 0.82 |
| | 18 | 50 | 4.08 ± 3.13 | 1.19 ± 0.47 | 1.50 ± 0.62 |
| | | 100 | 2.96 ± 1.84 | 1.02 ± 0.59 | 2.23 ± 1.08 |
| 25 | 6 | 50 | 1.76 ± 1.31 | 0.93 ± 0.47 | 1.37 ± 0.95 |
| | | 100 | 1.54 ± 1.55 | 1.35 ± 1.75 | 2.65 ± 5.34 |
| | | 150 | 1.69 ± 1.30 | 1.39 ± 1.49 | 1.59 ± 2.87 |
| | 12 | 50 | 3.99 ± 6.15 | 1.13 ± 0.34 | 2.05 ± 5.01 |
| | | 100 | 2.52 ± 2.49 | 1.24 ± 1.41 | 1.25 ± 0.48 |
| | | 150 | 1.47 ± 0.80 | 1.27 ± 1.20 | 1.03 ± 0.59 |
| | 18 | 50 | 5.52 ± 10.17 | 1.19 ± 0.73 | 1.21 ± 0.47 |
| | | 100 | 4.02 ± 11.07 | 1.19 ± 0.82 | 1.43 ± 1.68 |
| | | 150 | 2.77 ± 6.11 | 1.13 ± 0.59 | 1.22 ± 0.83 |
| 30 | 6 | 50 | 2.22 ± 2.98 | 2.09 ± 6.19 | 0.94 ± 0.27 |
| | | 100 | 1.97 ± 1.47 | 1.15 ± 1.19 | 1.04 ± 0.33 |
| | 12 | 50 | 1.44 ± 0.67 | 0.85 ± 0.32 | 1.08 ± 0.24 |
| | | 100 | 2.16 ± 2.81 | 1.43 ± 2.34 | 1.20 ± 0.91 |
| | 18 | 50 | 1.46 ± 0.80 | 0.94 ± 0.60 | 1.46 ± 1.94 |
| | | 100 | 1.75 ± 0.92 | 0.97 ± 0.30 | 1.47 ± 2.34 |

Studies on *Artemia* from Great Salt Lake (Utah, USA) (*Provasoli & Pintner, 1980*; *Nambu, Tanaka & Nambu, 2004*), San Francisco Bay (*Berthélémy-Okazaki & Hedgecock, 1987*), Daqinghe Saltern (Hebei, China) (*Huang, Chen & Liu, 2001*), Ebinur Lake (Xinjiang, China) (*He et al., 2016*), and Barkol Salt Lake (*Wang, Asem & Sun, 2017*) showed that oviparity rates increased under short daylight. The present results support this. In most T-S treatments, the percentage of oviparous broods of *A. sinica* increased with decreasing daylight. However, when the temperature was 30 °C, the highest oviparity rate occurred in the L18 treatment (Table 2). *Berthélémy-Okazaki & Hedgecock (1987)* also found that short daylight has no effect on oviparity in the San Francisco Bay population at higher temperatures.

*Jia et al. (1995a)* reported that the oviparity rate of *A. sinica* increased with salinity (T26, "natural light", S10–300). Under similar conditions (T25, L12), the present results showed that oviparity rates also increased with salinity within the 100–200 PSU salinity range, but a maximum value was recorded at 50 PSU (Table 2). When the salinity was ≥100 PSU, the oviparity rate was markedly lower than those of *Jia et al. (1995a)* (S100: 21.3% *vs.* 53%; S150 (140 or 160): 26.0% *vs.* 87% or 88%; S200: 29.2% *vs.* 96%). These differences may be related to light condition (fixed light intensity and photoperiod *vs.* "natural light"),

food type (mixed diet *vs.* live *Dunaliella*; feeding yeast may induce oviparity (*Dutrieu, 1960*)), and food quantity (*Zheng et al., 2019*; *Zhang et al., 2020*). Additionally, it can be connected to the fact that *Jia et al. (1995a)* only collected data from the first brood. A study by *Browne & Wanigasekera (2000)* on the same population also showed higher oviparity rates at higher salinities, with a 2.8 times higher oviparity rate at 180 PSU than at 120 PSU. Studies on other *Artemia* species or populations showed that the effects of salinity on reproductive mode varied among species or populations. For example, as salinity increased, oviparity rates rose followed by a decline in the parthenogenetic *Artemia* populations from Iran (*Aalamifar et al., 2014*), Putian Saltern (Fujian, China), and Ebinur Lake (*Bian, 1990*). Oviparity rates increased in the parthenogenetic *Artemia* populations from Greece (*Abatzopoulos et al., 2003*) and Bohai Bay (China) (*Sui et al., 2013*). For *Artemia* of Mona Lake (California, USA) (*Dana & Lenz, 1986*), San Francisco Bay (*Berthélémy-Okazaki & Hedgecock, 1987*; *Sui et al., 2013*) and Dongfang Saltern (Hainan, China) (*Bian, 1990*), oviparity rates decreased. However, in the Vietnam population of *Artemia franciscana* (*Sui et al., 2013*) and the *Artemia urmiana* from Urmia Lake (Iran) (*Agh et al., 2008*), oviparity rates first decreased and then increased. It has been noted that ovoviviparity predominates in permanent lakes, while oviparity is associated with unstable, stressful habitats (*Gajardo et al., 2002*). Some *Artemia* populations inhabiting seasonally dried salt lakes have almost no ovoviviparity, as exemplified by those in Larnaca Salt Lake (*Browne & Wanigasekera, 2000*), Hoh Salt Lake (*Bian, 1990*), and Caka Salt Lake (*Chang, Asem & Sun, 2017*). The formation of resting eggs facilitates *Artemia* offspring to escape from extreme salinity, which may be an important survival strategy for *Artemia* in salt lakes with dramatic salinity fluctuations. However, there are differences in *Artemia* living in habitats with continuously high but tolerable salinities, such as the Great Salt Lake. Since resting eggs are unable to hatch at high salinities (the maximum hatching salinity recorded is 140 g/L; (*Thun & Starrett, 1987*)), the only way to take advantage of otherwise favorable conditions is through ovoviviparous reproduction (*Drinkwater & Clegg, 1991*). Interestingly, *Bian (1990)* reported that *Artemia* from brine pools of different salinities in Chengkou Saltern (Shandong, China) responded differently to salinity. The oviparity rate of *Artemia* from a 14.8° Be brine pond increased with salinity; *Artemia* from an 18.0° Be brine pond was almost completely oviparous at all salinities; and the oviparity rate of *Artemia* from a 22.0° Be brine pond decreased at the highest experimental salinity (experimental salinity range 3.4–18° Be). Whether this was due to different parthenogenetic strains with different genetic bases is unknown.

According to *Wang & Zhang (1995)*, *A. sinica* from Yuncheng Salt Lake was oviparous at 20 °C and 28 °C, and ovoviviparity occurred only at 15 °C (L24, S90). Under similar photoperiod and salinity conditions (L18, S100), the percentage of oviparous broods of *A. sinica* in the present experiments increased with temperature, but still had about 2/3 and 1/4 of ovoviviparous broods at 25 °C and 30 °C, respectively (Table 2). This is remarkably different from the results of *Wang & Zhang (1995)*. *Wang & Zhang (1995)* cultured *Artemia* under continuous light and fed with *Dunaliella salina*, while *A. sinica* tends to be oviparous under a combination of longer daylight and higher temperature and when fed with live algae (see above). Other studies showed that the effects of temperature on reproductive mode varied among species and populations. For instance, *Artemia* of Great Salt Lake

had a higher oviparity rate at 28 °C (*Nambu, Tanaka & Nambu, 2004*). *Artemia salina* was almost entirely oviparous at 24 °C and 30 °C (*Browne, Davis & Sallee, 1988*). *Artemia* from San Francisco Bay reached the maximum oviparity rate at 24 °C (*Browne, Davis & Sallee, 1988*). *Artemia persimilis* from Hidalgo (Argentina) was all ovoviviparous at 15 °C (*Browne & Wanigasekera, 2000*). The oviparity rate of parthenogenetic *Artemia* from Greece was the lowest at 30 °C (*Abatzopoulos et al., 2003*). The parthenogenetic *Artemia* from Barkol Salt Lake tended to produce resting eggs under low-temperature conditions (*Wang, Asem & Sun, 2017*).

Little is known about how environmental factors affect the reproductive mode or diapause of *Artemia*. Studies on other arthropods have suggested that external environmental factors might control animal diapause in different ways, through direct induction and by influencing the animals' growth, development, and metabolism (*e.g.*, *Clay & Venard, 1972*; *Danks, 1987*; *Hou & Hua, 2003*). Photoperiod usually plays a leading role in the induction of arthropod diapause (*e.g.*, *Stross, 1966*; *March, 1982*; *Danks, 1987*; *Alekseev, Hwang & Tseng, 2006*). *Danks (1987)* suggested that salinity, humidity, food, and various chemical factors might be by influencing the development or metabolism of animals (*e.g.*, fat reserves and synthesis of photoreceptor pigments) to influence the effective sensing of photoperiod and thus the occurrence of diapause. Under natural conditions, insects sensed photoperiodic information through photoreceptors and transmitted it to the photoperiodic clock, which measured the information about long or short light exposure and transmitted it to the regulatory system. The insects then underwent a series of changes at the level of regulated genes and proteins that caused the insects to diapause (*Hori et al., 2014*). A similar mechanism may exist in *Artemia*, whose reproductive modes are also determined predominantly by photoperiod (*Wang, Asem & Sun, 2017*).

*Dana & Lenz (1986)* reported that the oviparity rate of the Mona Lake *Artemia* population increased with the brood number. In a study on 12 populations, *Browne et al. (1984)* found that the relationship between brood number and oviparity rate varied among species or populations. However, most populations showed a decreasing trend during the late reproductive period. *Berthélémy-Okazaki & Hedgecock (1987)* reported that the oviparity rate increased in the first seven broods for *Artemia* from San Francisco Bay, similar to a trend reported by *Browne et al. (1984)*. The oviparity rates of *A. sinica* in most treatments fluctuated greatly during the reproductive period, with some treatments (especially the 30 °C treatments) showing a decreasing trend, and only a few treatments (*e.g.*, T16-L12-S50) showing an increasing trend (Figs. 4A–4C). These results suggest that the relationship between oviparity rate and brood number is complex, but older females of *Artemia* may tend to reproduce ovoviviparously.

Although environmental conditions play a crucial role in determining the reproductive mode of *Artemia*, different species or populations seem to have different preferences. For instance, the *A. franciscana* from Grassmere Lake (New Zealand) is predominantly ovoviviparous (*Wear, Haslett & Alexander, 1986*). The parthenogenetic *Artemia* from Bohai Bay and *A. franciscana* from San Francisco Bay tend to be ovoviviparous (*Triantaphyllidis et al., 1995*). *Browne & Wanigasekera (2000)* reported that *A. salina* produced only resting eggs under all experimental conditions, while *A. sinica*, *A. persimilis,* and *A. franciscana*

were predominantly ovoviviparous. The present study showed that *A. sinica* was inclined to be ovoviviparous, which was similar to the result of *Browne & Wanigasekera (2000)*, but different from that of *Wang & Zhang (1995)* (see above).

## Lifespan and fecundity

Lifespan and reproductive period are mainly influenced by temperature. The lifespan of reproductive females is negatively correlated with temperature in the present study, which is in accordance with that observed in *A. franciscana*, *A. salina,* and some parthenogenetic *Artemia* populations (*Von Hentig, 1971*; *Browne, Davis & Sallee, 1988*; *Jia et al., 2002*). The reproductive periods of *A. sinica*, *A. salina*, *A. franciscana*, *A. persimilis*, and parthenogenetic *Artemia* were shorter at 15 °C and 30 °C than at 24 °C, and they constituted the majority of the lifespan at 24 °C (*Browne & Wanigasekera, 2000*). The present results also showed that the reproductive period was the longest at the moderate temperature (25 °C) and accounted for the largest proportion of the lifespan (Table 2; Fig. 1). The pre-reproductive period tends to be shortened with increasing temperature (*Browne, Davis & Sallee, 1988*; *Browne & Wanigasekera, 2000*; *Abatzopoulos et al., 2003*), which is supported by the present study. *Browne & Wanigasekera (2000)* suggested that the pre-reproductive period of *A. sinica* was shorter than other *Artemia* species. *Wang & Zhang (1995)* found that the pre-reproductive period of the Yuncheng population was longer at 15 °C, but shorter at 28 °C than most other *Artemia* populations. In the present study, the pre-reproductive period of *A. sinica* differed greatly at high and low temperatures. The maximum value (66.9 ± 7.8 d) observed in a low-temperature treatment (T16-L18-S100) was 4.2 times higher than the minimum value (16.1 ± 1.8 d) in a high-temperature treatment (T30-L18-S50). Regarding the effect of salinity, the pre-reproductive period of *A. sinica* was the shortest at median salinity (100 PSU) when the temperature was 25 °C, while it was the shortest at lower salinity (50 PSU) when the temperature was 16 °C, and showed no significant among salinity difference when the temperature was 30 °C (Table 2). In other studies, the pre-reproductive periods of parthenogenetic *Artemia* from Greece (*Abatzopoulos et al., 2003*) and Bohai Bay (*Wang & Zhang, 1995*; *Sui et al., 2013*), *Artemia monica* from Mona Lake (*Dana & Lenz, 1986*) and *A. urmiana* from Urmia Lake (*Agh et al., 2008*) were prolonged under high salinities. The increased energy consumption for osmotic regulation is apparently the reason for the slow growth and delayed reproduction under higher salinities (*Dana & Lenz, 1986*; *He et al., 2001*). However, salinity did not significantly affect the pre-reproductive periods of the San Francisco Bay and Vietnam populations of *A. franciscana* (*Sui et al., 2013*). Therefore, there are also interspecific and interpopulation differences in the effect of salinity on the maturation of *Artemia*.

For reproductive interval, previous studies showed that temperature and salinity had an interactive effect (*Wear, Haslett & Alexander, 1986*; *Browne & Wanigasekera, 2000*), with temperature having a greater effect than salinity (*Wear, Haslett & Alexander, 1986*), while photoperiod had no significant effect (*Huang, Chen & Liu, 2001*). The reproductive interval was often shorter at higher temperatures (*Browne, Davis & Sallee, 1988*; *Abatzopoulos et al., 2003*), but the effects of salinity varied among studies (*Triantaphyllidis et al., 1995*; *Abatzopoulos et al., 2003*; *Sui et al., 2013*). The present results showed that the interval was
independently affected by temperature and salinity, but there was no interaction between temperature and salinity (Table S1). The reproductive interval was reduced by higher temperature and lower salinity, with the effect of temperature being more pronounced (Table 2). In addition, although the products of oviparity are only embryos developing to the gastrula stage (*Benesch, 1969*; *Wang & Sun, 2007*), the length of the oviparous interval is slightly greater than that of the ovoviviparous interval (Fig. 2). The dry weight of the eggshell can be 22% of the weight of the entire egg (*Clegg et al., 1962*; *Von Hentig, 1971*). The production of resting eggs is more costly than that of nauplii (*Browne, 1980*). Therefore, constructing the multi-layered and delicate shell for oviparous eggs in the ovisac may be time-consuming.

Studies have shown that the fecundity of *Artemia* is the highest at moderate temperatures (*Browne, Davis & Sallee, 1988*; *Vanhaecke & Sorgeloos, 1989*; *Jia et al., 1995b*; *Browne & Wanigasekera, 2000*), and decreases with increasing salinity (*Dana & Lenz, 1986*; *Triantaphyllidis et al., 1995*; *Wang & Zhang, 1995*; *Jia et al., 1995a*; *Agh et al., 2008*; *Sui et al., 2013*). Similar results were obtained in the present study, which revealed that salinity was the main influencing factor of *A. sinica* fecundity. The low fecundity under high salinity may be related to *Artemia* consuming additional energy for osmotic regulation, and less energy is available for reproduction (*Dana & Lenz, 1986*). In the experiments of *Browne & Wanigasekera (2000)*, *A. sinica* failed to reproduce at 60 PSU. In the present study, however, all fecundity parameters achieved maximum values at 50 PSU. Some unknown environmental or biological factors may account for this difference. In addition, the present results also showed that the number of broods reached a maximum at a moderate temperature (25 °C), but the offspring per day increased with increasing temperature, which is apparently connected to quick metabolism and development under higher temperatures.

*Dana & Lenz (1986)* reported that the brood size of *Artemia* from Mona Lake increased with brood number (they only provided data for the first three broods). *Alexander* (*1982*, as cited in *Wear, Haslett & Alexander, 1986*) found that the number of nauplii per brood was higher at the middle, but lower at the beginning and end of the reproductive period. *Browne et al. (1984)* showed that the reproductive output of almost all of the 12 studied populations declined at the end of the reproductive period, with the only exception (the Santa Pola population, Spain) believed to be a mixture of parthenogenetic and bisexual *Artemia*. These authors concluded that *Artemia* had little energy reserves after a high reproductive output, leading to high mortality after the reproductive period. However, the brood size of *A. sinica* in our study kept increasing throughout the reproductive period and did not decline even in the last two broods (Fig. 4; Table 3). It was observed that most females were still holding eggs at the time of death, and the length of the post-reproductive period was less than the reproductive interval (Table 2). Hence, *A. sinica* seems still in the reproductive period even before death, and there is no "decline" in the reproductive capacity. In addition, for the same brood number of *A. sinica*, there is no significant difference between the sizes of oviparous and ovoviviparous broods (Table S2), which is consistent with that observed in the Mona Lake population (*Dana & Lenz, 1986*). Oocytes of *Artemia* mature in batches, and the destiny (to become nauplii or resting eggs) of each batch of oocytes is determined by the token stimuli signals when the oocytes develop to

the previtellogenesis stage (*Wang et al., 2019*). This can explain the similarity of the brood sizes between oviparity and ovoviviparity, though the production of resting eggs is more costly than that of nauplii (see above).

*Browne et al. (1984)* documented that the correlation between the pre-reproductive period and other reproductive parameters was extremely low, while the reproductive interval was positively correlated with the number of broods, total offspring, and female lifespan. For *A. sinica*, the pre-reproductive period and the reproductive interval are not significantly correlated with other parameters (Table S3). Since the analysis of *Browne et al. (1984)* was based on mixed data from 12 *Artemia* populations, some reproductive traits (*e.g.*, total offspring, offspring per day, and reproductive interval) might be influenced by genetics. Besides, strong positive correlations are found among reproductive period, number of broods, and total offspring (Table S3), coinciding with that reported by *Browne, Davis & Sallee (1988)*.

Table S4 compares reproductive parameters of different *Artemia* populations under "optimal" conditions. For *A. sinica* from Yuncheng Salt Lake, the oviparous rate (21.3%) is low (ranking: 25/38), the total offspring, offspring per brood, and offspring per reproductive day are moderate (ranking: 15/31, 13/31, 16/33, respectively), and the pre-reproductive period is short (ranking: 5/38). Since the population growth rate of zooplanktons is influenced most by the pre-reproductive period and less by the brood size and lifespan (*Allan, 1976*), these *A. sinica* characteristics make it one of the most capable *Artemia* in terms of population growth (replenishment). This may be the major reason that the commercial production of *Artemia* at Yuncheng Salt Lake is dominated by harvesting live adults rather than resting eggs (*Jing, 2020*). The high population growth is further supported by the fact that *A. sinica* has become a successful invasive species in some coastal salterns (*Van Stappen et al., 2007*).

## CONCLUSIONS

All reproductive parameters of *A. sinica* can be influenced at least by one of the three environmental factors examined (temperature, photoperiod, and salinity), and environmental factors often have interactive effects on *A. sinica* reproduction. With the elevation of temperature, the lifespan, the pre-reproductive period, and the reproductive interval decrease, while the number of offspring per day increases. *Artemia sinica* has the maximum number of broods and the longest reproductive period at a moderate temperature (25 °C). The total offspring, the brood size, and the offspring per day are negatively correlated with salinity. *Artemia sinica* tends to reproduce oviparously under low temperatures and short daylight conditions, and ovoviviparously under high temperatures and long daylight conditions. The oviparous interval is often longer than the ovoviviparous interval. *Artemia sinica* remains reproducing until death, and its reproductive capacity (brood size) keeps increasing during its lifetime. Compared with other *Artemia* species or populations, *A. sinica* from Yuncheng Salt Lake has a relatively shorter pre-reproductive development time, a preference for ovoviviparity, and a relatively higher fecundity and population growth capacity, making it a suitable culture species for obtaining fresh biomass.

## ACKNOWLEDGEMENTS

We are grateful to Professor Gilbert Van Stappen for kindly providing the *Artemia* resting eggs. We thank Mr. Liqun Guo for his help in experiments.

### Funding

This work was supported by the Fundamental Research Funds for the Central Universities (Ocean University of China) (No. 202164001). The funders had no role in study design, data collection and analysis, decision to publish, or preparation of the manuscript.

### Grant Disclosures

The following grant information was disclosed by the authors:
The Fundamental Research Funds for the Central Universities (Ocean University of China): 202164001.

### Competing Interests

The authors declare there are no competing interests.

### Author Contributions

- Jing-Yu Yang conceived and designed the experiments, performed the experiments, analyzed the data, prepared figures and/or tables, authored or reviewed drafts of the article, and approved the final draft.
- Shi-Chun Sun conceived and designed the experiments, authored or reviewed drafts of the article, and approved the final draft.

### Data Availability

The raw data are available in the Supplementary Files.

### Supplemental Information

Supplemental information for this article can be found online at http://dx.doi.org/10.7717/peerj.15945#supplemental-information.

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
