# Peer review of "Combined effects of temperature, photoperiod, and salinity on reproduction of the brine shrimp Artemia sinica (Crustacea: Anostraca)"

_PeerJ, doi:10.7717/peerj.15945_

## Round 0.1 · original submission · Major Revisions

I agree with the reviewers that albeit the manuscript contains interesting results and the research framework is sound, it needs major revision, especially in terms of language consistency. Professional proofreading of the revised manuscript is highly recommended. Also, the discussion section lacks depth and is too long.

Reviewer 1 ·

Basic reporting

General comments:

1. Overall, the manuscript is well written and structured, and includes relevant literature, figures and tables.
2. Clear and unambiguous English used throughout the manuscript. However, in some parts of the manuscript, long sentences and phrasing makes comprehension difficult. I have listed some comments below that need to be addressed.
3. Title of each section (e.g. Introduction; Materials and Methods) should be written in lowercase letters.

Abstract:

Line 36: “50/100 psu”. Did you mean “50 or 100 psu”? Try to avoid using slash symbol as it can add confusion in your writing.
Line 38: “species/populations”. Same as comment for Line 36.

Introduction:

Line 56-57: “Liang and MacRae, 1999” should be written “Liang & MacRae, 1999”. Please make necessary changes throughout the manuscript.
Line 87: “Cai, 1989”. Is this a citation?
Line 92: “A. sinica”. It is better to spell out the genus in full at the beginning of a sentence.
Line 94: I suggest rephrasing “natural population of Yuncheng Salt Lake” to “natural population of A. sinica in Yuncheng Salt Lake”.
Line 97-102: “For example, Wang and Zhang (1995) found that A. sinica was oviparous at 20°C and 28°C, and ovoviviparous only at 15°C, while Browne and Wanigasekera (2000) reported only 22.9% of oviparous offspring at 30°C; Jia et al. (1995a) reported that A. sinica reproduced successfully at the salinity range of 10-300 psu, and the proportion of oviparity increased with increasing salinity, whereas Browne and Wanigasekera (2000) reported that A. sinica did not survive to reproduction under a salinity of 60 psu”. This sentence is too long and the current phrasing makes the sentence difficult to comprehend. I suggest separate the findings on temperature and salinity into two separate sentences.
Line 111: I suggest rephrasing “in this paper” to “in this study”.

Materials and Methods:

Line 114: “Artemia sinica”. The abbreviation “A. sinica” should be used consistently throughout the manuscript.
Line 181-182: Change “% Ovoviviparous offspring: percentage of resting eggs in “total offspring” for an effective female” to “% Ovoviviparous offspring: percentage of nauplii in “total offspring” for an effective female”.
Line 188: I suggest rephrasing “SPSS26” to “SPSS 26” or “SPSS version 26”.

Results:

Line 192: “Tab. 1” should be written in full and non-italic (Table 1).
Line 197: “Tab. 2”. Same as comment for Line 192. Please make necessary changes throughout the manuscript.
Line 225: “Fig. 1” should be written in non-italic (Fig. 1).
Line 234: “Fig. 2”. Same as comment for Line 225. Please make necessary changes throughout the manuscript.

Discussion:

Line 307-311: “Our results are similar to that of Wang et al. (2017), who reported interactive effects of temperature, photoperiod and salinity on the reproductive mode of the parthenogenetic Artemia (2n clones) from Barkol Salt Lake (Xinjiang, China), lower temperature and shorter daylight tending to induce production of resting eggs, but salinity without significant independent effect on reproductive mode”. This sentence is too long and the phrasing is difficult to comprehend. I suggest rephrase this sentence into two separate sentences.
Line 326-327: ‘’but which were lower than those at 50 psu’’. Please rephrase the sentence.
Line 329-333: “These differences may be related to light condition (fixed light intensity and photoperiod vs “natural light”), food type (mixed diet vs live Dunaliella; feeding yeast may induce oviparity (Dutrieu 1960 as cited in Berthélémy-Okazaki and Hedgecock, 1987)) and quantity (Zheng et al., 2019; Zhang et al., 2020), and may also be related to that Jia et al. (1995a) collected data only from the first brood”. Please rephrase as the sentence is too long and difficult to comprehend.
Line 389: I suggest rephrasing “a trend like that of Browne et al. (1984)” to “similar to a trend reported by Browne et al. (1984)”.
Line 410-411: “Female lifespan of A. sinica”. Please rephrase the sentence.
Line 435: “interspecific/interpopulation”. Please write in full and try to avoid using slash symbol as it can add confusion in your writing.
Line 436: “maturation speed”. I suggest remove “speed” from the sentence.
Line 437-440: I suggest rephrasing “For reproductive interval, previous studies have shown that temperature and salinity had an interactive effect (Wear et al., 1986; Browne and Wanigasekera, 2000) and temperature had a greater effect than salinity (Wear et al., 1986), photoperiod had no significant effect (Huang et al., 2001)” to “For reproductive interval, previous studies have shown that temperature and salinity had an interactive effect (Wear et al., 1986; Browne and Wanigasekera, 2000), with temperature had a greater effect than salinity (Wear et al., 1986), while photoperiod had no significant effect (Huang et al., 2001)”.
Line 448-451: “This may be related to that the production of resting eggs is more costly than that of nauplii (Browne, 1980) (the dry weight of the eggshell can be 22% of the weight of the entire egg (Clegg et al., 1962; Von Hentig et al., 1971)), and the construction of the multi-layered, delicate shell for oviparous eggs in the ovisac may be time consuming”. Please revise as the phrasing is difficult to comprehend.
Line 456: I suggest rephrasing “Similar results are obtained in the present study” to “Similar results were obtained in the present study”.
Line 462: “environmental/biological”. Same as comment for Line 435. Please make necessary changes throughout the manuscript.
Line 477: I suggest rephrasing “immediately before death” to “even before death”.
Line 498: I suggest rephrasing “catching fresh adults” to “harvesting live adults”.
Line 500-501: “For example, A. sinica has recently become the dominant species in saltpans in Tanggu and Hangu, China (authors’ observation)”. Please avoid citing an unpublished data.

Conclusions:

Line 507-508: “Temperature, photoperiod, salinity, and their interactions have significant effects on the reproduction of A. sinica”. Did you mean interactions of all environmental factors or only certain factors? Please revise and try to avoid an ambiguous sentence.
Line 509-510: “Artemia sinica”. The abbreviation “A. sinica” should be used consistently throughout the manuscript.
Line 515: “Artemia sinica seems to be still in the reproductive period immediately before death”. Avoid repetition in your writing. Please rephrase the sentence.

Experimental design

The authors have presented an original research of the combined effects of environmental factors on reproductive capacity of A. sinica from Yuncheng Salt Lake, China. The experimental design is well defined, relevant and meaningful. However, I have some concerns regarding the salinity used in this study (50, 100, 150, and 200 psu). Salinity 100 psu and above are considered high salinity for Artemia. A high salinity can be stressful to this animal, thus resulted in extra energy used for osmotic regulation and less energy available for reproduction. Authors should consider using a broad range of salinity for the experiments, i.e. low, moderate and high. I believe this will provides more precise results on the effects of salinity on reproduction of Artemia.

Validity of the findings

The rational and significance of the experiments are clearly stated, and the results are statistically sound. However, I noticed that some data for salinity 150 and 200 psu were not listed in Table 3. Please confirm whether the data are available or not. If not, I don’t think it is proper to make any conclusion related to the incomplete data set. For example, “The total offspring, offspring per brood, offspring per day and offspring per reproductive day all decreased with the elevation of salinity (Table 3) (Line 241-242).

Reviewer 2 ·

Basic reporting

1) The manuscript consists of several interesting and important findings featuring the combined effects of temperature, salinity and photoperiod towards the reproduction mode and performance of the brine shrimp Artemia sinica. Nevertheless the paper is generally not so well structured. Editing by a fluent English speaker is strongly recommended because grammatical errors are scattered throughout the paper.

2) In my opinion, the description of the ANOVA analysis results (Line 197-211) and Table 2 are unnecessary because the significant differences can be incorporated in the figures/figure legends.

3) Sentence from line 230-231 requires revision

4) In Line 235 and 238, Higher "numbers" should be Higher "number"

5) The section on reproductive modes needs to be revised. The authors used the phrase "higher percentage" a number of times, but they never explained what the percentages meant.

6) The discussion lacked organization. Without placing any attention on highlighting the potential reasons why the reproductive mode and other characteristics were affected by these exposures, the results were merely compared to the work of other coworkers. The authors should explain in the context of how these physical characteristics might have changed reproduction modes or performance. There could have some assumption included and discussed.

Experimental design

The methodology used in this study is sound, with sufficient details and information given.

Validity of the findings

The findings are novel, and all supporting information has been provided. However, the findings were poorly phrased and presented. The general presentation of the results and discussion part can be enhanced by thorough review and editing by a Native speaker. The current version must be revised.

---

## Round 0.2 · Minor Revisions

The manuscript is almost ready for publication. Kindly address the reviewer's comments and I look forward to reading the final version of the manuscript.

Reviewer 1 ·

Basic reporting

Overall, the manuscript has been substantially improved following the first revision. All comments have been addressed by the authors. However, on reading the revised manuscript, some minor corrections are required:

Line 253: Change “30ºC treatments” to “30ºC treatment”.

Line 282-284: I suggest rephrasing “Their results showed that lower temperatures and shorter daylight induce the production of resting eggs, while salinity did not have a significant independent effect on reproductive mode. Our results are similar to this.” to “Similar to this study, their results showed that lower temperatures and shorter daylight induce the production of resting eggs, while salinity did not has a significant independent effect on reproductive mode.”

Line 284-287: I suggest rephrasing “Berthèlèmy-Okazaki & Hedgecock (1987) found photoperiod-temperature and temperature-salinity had interactions affecting the reproductive mode of Artemia from San Francisco Bay (California, USA), but did not detect a significant interaction between photoperiod and salinity.” to “Berthèlèmy-Okazaki & Hedgecock (1987) found that photoperiod-temperature and temperature-salinity had interactions affecting the reproductive mode of Artemia from San Francisco Bay (California, USA), but did not detect a significant interaction between photoperiod and salinity.”

Line 294-296: I suggest rephrasing “Berthèlèmy-Okazaki & Hedgecock (1987) also found that the induction effect of short daylight on oviparity in the San Francisco Bay population disappeared at higher temperatures.” to “Berthèlèmy-Okazaki & Hedgecock (1987) also found that short daylight has no effect on oviparity in the San Francisco Bay population at higher temperatures.”

Line 308-310: I suggest rephrasing “Studies on other Artemia species or populations showed that the salinity effects on reproductive mode varied among species or populations.” to “Studies on other Artemia species or populations showed that the effects of salinity on reproductive mode varied among species or populations.”

Line 326-329: I suggest rephrasing “Since resting eggs cannot hatch at high salinities (the maximum hatching salinity recorded is 159 g/L; Dana & Lenz, 1986), the only way to take advantage of otherwise favorable conditions is through ovoviviparous reproduction (Drinkwater & Clegg, 1991).” to “Since resting eggs are unable to hatch at high salinities (the maximum hatching salinity recorded is 159 g/L; Dana & Lenz, 1986), the only way to take advantage of otherwise favorable conditions is through ovoviviparous reproduction (Drinkwater & Clegg, 1991).”

Experimental design

No further comment. The authors have addressed the comments in regards to the salinity used in this study and clarified the rational of the experimental design.

Validity of the findings

No further comment. The statistical analyses used in this study are appropriate. Relevant data are reported and adequately discussed.

---

## Round 0.3 · accepted · Accept

I applaud the authors for following through with the reviewing process and addressing all concerns and comments from the reviewers patiently. I look forward to reading the published version of this interesting paper on the reproduction aspects of brine shrimp!